# Formulation Study of a Co-Processed, Rice Starch-Based, All-in-One Excipient for Direct Compression Using the SeDeM-ODT Expert System

**DOI:** 10.3390/ph14101047

**Published:** 2021-10-14

**Authors:** Karnkamol Trisopon, Nisit Kittipongpatana, Phanphen Wattanaarsakit, Ornanong Suwannapakul Kittipongpatana

**Affiliations:** 1Department of Pharmaceutical Sciences, Faculty of Pharmacy, Chiang Mai University, Chiang Mai 50200, Thailand; mickey_imm@hotmail.com (K.T.); nisit.k@cmu.ac.th (N.K.); 2Research Center for Development of Local Lanna Rice and Rice Products, Chiang Mai University, Chiang Mai 50200, Thailand; 3Department of Pharmaceutics and Industrial Pharmacy, Faculty of Pharmaceutical Science, Chulalongkorn University, Bangkok 10330, Thailand; phanphen.a@chula.ac.th

**Keywords:** direct compression, co-process, rice starch, spray drying, all-in-one excipient, SeDeM-ODT expert system

## Abstract

A co-processed, rice starch-based excipient (CS), previously developed and shown to exhibit good pharmaceutical properties, is investigated as an all-in-one excipient for direct compression (DC). An SeDeM-ODT expert system is applied to evaluate the formulation containing CS, in comparison with those containing the physical mixture and the commercial DC excipients. The results revealed that CS showed acceptable values in all six incidence factors of the SeDeM-ODT diagram. In addition, the comprehensive indices (IGC and IGCB) were higher than 5.0, which indicated that CS could be compressed with DC technique without additional blending with a disintegrant in tablet formulation. The formulation study suggested that CS can be diluted up to 60% in the formulation to compensate for unsatisfactory properties of paracetamol. At this percentage, CS-containing tablets exhibited narrow weight variation (1.5%), low friability (0.43%), acceptable drug content (98%), and rapid disintegration (10 s). The dissolution profile of CS displayed that more than 80% of the drug content was released within 2 min. The functionality of CS was comparable to that of high functionality excipient composite (HFEC), whereas other excipients were unsuccessful in formulating the tablets. These results indicated that CS was a suitable all-in-one excipient for application in DC of tablets.

## 1. Introduction

Nowadays, tablet remains the most commonly used dosage form on the pharmaceutical market. In recent years, at least half of the drugs approved by the U.S. Food and Drug Administration’s Center for Drug Evaluation and Research (CDER) have been in solid dosage form (54% in 2019, 53% in 2018, and 50% in 2017) [1,2,3]. Among the general tablet manufacturing methods, direct compression (DC) stands the most prominent due to the cost-effectiveness, simplicity, and high stability it offers to the drug products [4]. However, manufacturing without a granulation process and recent advancement in pharmaceutical tableting, such as continuous manufacturing and high-speed tableting machines, could limit this production method with the conventional excipients [5,6]. Simultaneously, it encourages the development of a new excipient to support tablet manufacturing under specific production conditions.

In the past few decades, the concept of a co-processed excipient has gained a lot of attention, as it is the simplest and fastest way to develop a new excipient without the requirement to study toxicity for regulatory approval [7]. In general, a co-processed excipient outperforms the physical mixtures in terms of pharmaceutical properties because it synergizes the good properties of each excipient [8,9]. Moreover, a previous study has reported that the co-processed excipient improved content uniformity of dosage unit since the excipients were intimately combined, thus reducing the segregation problem [8]. For these reasons, the co-processed excipient, which plays multiple roles in the formulation, could support tablet manufacturing with the DC technique.

In the previous study, we reported the ability of a novel rice starch-based, co-processed excipient (CS), prepared using a spray drying technique, as a multifunctional excipient for DC [9]. Native rice starch (RS) is commonly used as a pharmaceutical excipient, but its application in DC is limited due to the poor flowability and the low disintegration property [10]. Upon co-processing with cross-linked carboxymethyl starch (CCMS) and silicon dioxide, the functionality was improved. The CSs showed agglomerate spherical particles, thus improving the flowability of the material. The co-processing with CCMS, a superdisintegrant, provided rapid disintegration time of the co-processed tablet, while still preserving an adequate compressibility profile [11]. Nevertheless, CS needed more investigation to be applied in DC formulation.

In the past, pharmaceutical formulation development depended mostly on experiments and research experience, which were time-consuming and costly processes. The SeDeM expert system developed by Carreras et al. [12] could improve this process based on the concept of pharmaceutical quality by design (QbD), which is a systematic approach to create a new pharmaceutical product based on a quality risk management and sound, logical science [13]. This is a new galenic method that is used in tablet pre-formulation and formulation studies, particularly for the DC method. The powder characteristics of active pharmaceutical ingredients (APIs) and excipients are evaluated through the SeDeM diagram to determine whether the materials are suitable for DC manufacturing or need additional excipients for tableting. Recently, a SeDeM-ODT expert system has been developed to include disgregability factor, making it more advantageous than the old SeDeM method in the assessment of excipient properties [14].

In this work, the SeDeM-ODT expert system is applied for the formulation study of CS, in comparison with the physical mixture and commercial excipients, as a multifunctional excipient for DC technique. Paracetamol is used as a model drug due to the unsatisfactory powder characteristics, which contribute to the difficulty in DC production. The powder characteristics of the excipients and the API are determined, and the paracetamol tablets are then produced based on the calculated percentage of a corrective excipient obtained from the SeDeM-ODT expert system.

## 2. Results and Discussion

### 2.1. Optimization of Excipients and APIs Powder Using SeDeM-ODT Expert System

The API was evaluated using a SeDeM diagram, which provided the characteristics of excipient powder in 5 factors derived from 12 parameters. In contrast, excipients were determined with a SeDeM-ODT diagram that included the three disgregability parameters (Figure 1 and Table 1). These factors indicated the suitability of the excipient for DC formulation.

#### 2.1.1. Dimension Factor

The dimension radius value of starch-based excipients (CS, PMSS, PGS, and ALM) were in the acceptable range (>5). ALM showed the highest dimension value as it exhibited the highest density profile (Table 1 and Table 2). On the other hand, the microcrystalline cellulose (MCC)-based excipients, SMCC and HFEC, showed low bulk and tapped densities, resulting in low dimension value. The dimension values of starch-based excipients lied between the two MCC values. Paracetamol had a dimension value lower than 5, suggesting that it might need a densification process before tableting [15]. These parameters related to flowability and compressibility as they were used for calculation of some parameters, including inter-particular porosity, Carr index, and Hausner ratio. A low powder density facilitated tablet compressibility, as there was more space to deform during compaction [16], while high powder density promoted a die-filling process of tableting [17].

#### 2.1.2. Compressibility

The compressibility was determined based on the inter-particular porosity, inter-particular interaction, and cohesiveness of the powder. All excipients showed compressibility values higher than 5, except PGS and ALM. The brittle characteristic of ALM provided low strength tablets [18] (Table 1 and Table 2), while the high plasticity of MCC-based excipients (SMCC and HFEC) improved their compressibility profile [19]. SMCC showed the highest compressibility value, while the lowered compressibility of HFEC was due to the hydrophobicity of the incorporated lubricant, which limited particle bond formation [20]. For starch-based excipients, CS and PMSS showed no significant difference in compressibility factor values. However, the cohesive index of CS was 1.6 times greater than that of PMSS. This was possibly due to the presence of the amorphous CCMS and sodium silicate in the co-processed CS particles. The increase in amorphous content enhances material plasticity and, thus, improves compressibility [21]. PGS exhibited unacceptable compressibility incidence. The high density of PGS (Da of 5.42 and Dc of 7.09) indicated low particular porosity, which limited particle deformation [16]. Furthermore, PGS required a long dwell time to form a sufficient bond between the particles, resulting in low compressibility [10]. On the other hand, a very high elastic material such as paracetamol exhibited a low compressibility value (4.39), implying that it needed a filler-binder to compensate for compressibility [22].

#### 2.1.3. Flowability

All excipients showed good flowability (>6) suitable for the DC method (Table 1 and Table 2). HFEC exhibited the greatest flowability as a result of having a lubricant in the composite, while SMCC flowability was reportedly enhanced by silicification. The flowability of the spray-dried products (CS and ALM) was adequate, and no significant difference was observed between them. The spray dry technique produced agglomerate particles, which in turn promoted material flowability [23]. In contrast, PMSS was inferior to other excipients owing to the small particle size of RS, the main component. For PGS, the modification increased the particle size and yielded better flowability than PMSS (Table 1). Due to its very fine powder, paracetamol could not flow through the orifice of the funnel; thus, the angle of repose and powder flow could not be determined. The flow incidence value of paracetamol (2.65) indicated that DC was not the possible option for tablet production and other techniques, such as dry or wet granulations, should be employed [24]. Naturally, the angle of repose and the powder flow showed a good correlation in Pearson analysis (Appendix A), as both were consistent with the flow appearance of the powders. A positive correlation was also observed between angle of repose and Hausner ratio. On the other hand, the angle of repose had a significant negative correlation with the Carr index. This parameter demonstrated the particles interaction, which was inversely related to flowability.

#### 2.1.4. Stability

The stability factors of all excipients were above 5, indicating good future stability (Table 1 and Table 2). The hygroscopicity of the materials was influenced by powder characteristics, such as surface area, porosity, hydrophilic functional group, and crystallinity structure [21]. As all studied excipients were glucose-based materials, the difference in the bond formation between glucose units impacted the crystallinity of polymer, which in turn affected the moisture sorption property [25]. The crystalline solid absorbed water mostly on the particle surface, while the less dense, amorphous solid allowed water vapor molecules to penetrate the surface and formed bonds with those in the bulk phase. Thus, amorphous solid absorbed water more than the crystalline counterpart [26]. This explained why ALM, which possessed a crystalline structure, exhibited the lowest hygroscopicity (defined as slightly hygroscopic, with mass increase between 0.2–2% *w*/*w*). In contrast, the MCC-based and starch-based excipients (polysaccharides) were graded as moderately hygroscopic (mass increase of 2–15% *w*/*w*) [27]. The result showed that hygroscopicity corresponded with loss on drying (%), which represented the moisture content in materials. All excipient showed a loss on drying percentage within the USP specifications [28,29,30], while paracetamol showed a slight hygroscopicity due to the crystalline nature of the material [31]. The presence of moisture in materials promoted powder compressibility, which was confirmed by a significant Pearson correlation between the loss on drying and cohesive index (Appendix A).

#### 2.1.5. Dosage

This factor relates to the uniformity of the powder and the dosage of the finished product. Most excipients showed an acceptable value of dosage factor (equal or higher than 5), with the exception of HFEC (Table 1 and Table 2). CS and PMSS possessed high Iϴ values, suggesting low particle size distribution, thus promoting powder uniformity and reducing powder segregation during tablet production [32]. Conversely, ALM-, PGS-, and MCC-based excipients showed low Iϴ values, which may increase the risk of powder segregation. The Pearson analysis revealed that the size distribution inversely correlated with the %Pf. This value represents the percentage of powder that passes through the 50 μm sieve. Therefore, excipients with low %Pf commonly exhibited narrow size distribution (high Iϴ values). The suitable particle size for DC production should be between 100–200 μm in order to promote flowability and compressibility. Particles of smaller sizes than that range could inhibit the material flowability [33]. A higher %Pf value indicated a higher number of small particles, which negatively affected flowability. This effect can be observed in PMSS that exhibited unacceptable %Pf value, and showed the worst flowability compared to other excipients. Moreover, the irregular shape of RS particles could limit the free flowability among the PMSS particles. Low %Pf was also observed in CS, but did not negatively affect the flowability. This was because the spherical shape of CS agglomerates and the presence of silicon dioxide reduced the cohesive force between the particles, thus promoting the flowability. For paracetamol, it exhibited an adequate Iϴ value, while %Pf was lower than the acceptable value. This result indicated a narrow size distribution of paracetamol and that the particle size was low, as API is usually produced as a fine powder.

#### 2.1.6. Disgregability

The disgregability factor represents the ability of the excipient to undergo self-disintegration. The result revealed that CS, PGS, HFEC, and ALM showed desirable disgregability values (Table 1 and Table 2). As expected, the DE values showed a high correlation with DSD and DCD values, as these parameters corresponded with disintegration ability of excipients. This result implied that these excipients could provide fast-disintegrating tablets, which was appropriate for the formulation of orally disintegrating tablets (ODT). Among all excipients, CS showed the best disgregability factor, followed by HFEC, as they contained a disintegrant. PGS was partially gelatinized, causing swelling of starch granules, whereas ALM was a high-water soluble excipient, promoting disintegration [34,35]. On the other hand, PMSS and SMCC did not pass the acceptable value. The physical mixing with a high percentage of super-disintegrant (CCMS) limited tablet disintegration, as it created a viscous gel that inhibited water penetration into a tablet. Low disgregability was also observed in SMCC due to a strong bond formation, thus limiting disintegration of the tablet.

### 2.2. The Determination of the Indices Using the SeDeM-ODT Diagram

The suitability of API and excipients for the application in DC formulation was determined with the comprehensive indices that were calculated based on SeDeM and SeDeM-ODT diagrams, respectively (Table 2). According to the SeDeM diagram of paracetamol, most incidence factors were below 5, except the stability factor (9.74). The comprehensive indices showed unsatisfactory values, namely that the IPP (4.86) and IGC (4.63) values were in the range of 3–5. This result implied that it was not suitable for manufacturing with the DC method and required the additional excipient to improve the powder characteristics.

The comprehensive indices of excipients were calculated based on the SeDeM-ODT diagram. The results revealed that all excipients showed a comprehensive index (IP, IPP, and IGC values) higher than 5, indicating that they were suitable for DC of tablets. However, CS, PGS, ALM, and HFEC had IGCB values that surpassed 5 (6.15, 5.27, 6.18, and 6.29, respectively). This result implied that these excipients can be produced by the DC technique without the requirement of a disintegrant to formulate tablets and they can also be applied in ODT formulation. Moreover, CS was the only excipient that exhibited acceptable characteristics in all incidence factors (higher than 5). On the other hand, the IGCB values of SMCC (4.62) and PMSS (4.57) were lower than 5, suggesting that they required a disintegrant to formulate DC tablets [36].

### 2.3. The Correction of API Characteristics for DC Formulation

The SeDeM diagram of API was generally evaluated based on five factors. Among these, however, compressibility and flowability were considered the two most important factors which highly affected the success of tablet production. Therefore, the correction of paracetamol powder was focused on these factors. The amount of excipients required to adjust paracetamol compressibility and flowability (CP, %) were calculated, where the mean incidence radius value required to be corrected (R) was set to 5 to achieve an acceptable characteristic (Table 3). Once the values are obtained, the factor which requires a higher percentage of excipients (shows higher %CP) between the two is chosen.

To correct paracetamol compressibility, SMCC showed the best dilution potential. It required 25% of SMCC to compensate for paracetamol compressibility, followed by HFEC (30%), CS (45%), and PMSS (55%) (Table 3). PGS and ALM were not selected to adjust for paracetamol compressibility, as both showed unacceptable compressibility values. However, the DC formulation also required a suitable flowability for tableting. The result suggested that paracetamol required 52% of HFEC to correct flowability, while SMCC, CS, ALM, and PGS were required at 58%, 60%, 63%, and 65% in the formulation, respectively (Table 3). PMSS, which showed the worst flowability, was required at 70% addition. From the data, the %CP to correct paracetamol characteristics was selected based primarily on the flowability, as a higher number of excipients was required compared to compressibility.

Using the calculated CP value for characteristic compensation, the paracetamol formulations, prepared as powder blends, were analyzed using the SeDeM-ODT diagram (Figure 2). All formulations showed acceptable IP (0.58–0.75), IPP (5.43–5.92), and IGC (5.17–5.64) values, indicating their suitability for DC manufacturing. Most formulations showed compressibility factors higher than 5, with the exception of PGS and ALM formulations (Table 4), which were hard to consolidate. The flowability result revealed that all formulations showed an improvement in flowability. CS, PGS, SMCC, and ALM formulations showed no significant difference in flowability, while the HFEC formulation exhibited the lowest flow value, as it was rather diluted with paracetamol. However, the result revealed that the flowability value of all formulations was lower than the acceptable radius value (2.97–4.55 of flowability). The SeDeM-ODT diagrams revealed that the CS, PGS, SMCC, and HFEC showed satisfactory IPP and IGCB values, implying that these formulations possessed the ability to be compressed by the DC method without the addition of disintegrant.

As this expert system estimated powder behavior of materials based on the linearity assumption, the flowability values of formulation should be close to the estimated R value (5.00). However, all formulations showed flow values lower than the estimated R value, indicating that the characteristics of powder mixture may not follow a linear system [37].

### 2.4. Formulation Study

The formulations were prepared using the ratios obtained from the CP values. At these percentages, CS, PMSS, and SMCC were successful in producing paracetamol formulations. These formulations showed satisfactory tablet properties (Table 5). They showed low weight variation within the acceptance value. The friability was lower than 1.0%, at which the drug content was within the acceptable range (95–105%). Among the successful formulations, CS showed the best disintegration property, namely seven times faster than PMSS and SMCC formulations. On the other hand, PGS, HFEC, and ALM failed to produce paracetamol tablets. High tablet friability was observed in these formulations. Moreover, PGS and ALM formulations showed a high weight variation and low drug content, indicating that the number of excipients was not sufficient to achieve tablet requirements. However, most formulations, except those contained PGS and ALM, had adequate flowability, which represented a low weight variation, even though the flowability value from the SeDeM diagram was lower than 5. This result suggested that the calculated CP (%) value may not be precise due to the non-linearity of the powder blend.

As the SeDeM expert system was studied based on 12 parameters, the reliability factor was 0.952. Thus, the variation of calculated CP (%) should be 5%. Therefore, the excipient percentage was varied by 5% from the CP value to produce paracetamol formulations. For successful formulations (CS, PMSS, and SMCC), the number of excipients was decreased by 5% to formulate tablets. The results showed that all formulations failed to produce tablets. CS formulation exhibited high tablet friability, while PMSS and SMCC formulations showed high weight variation, implying insufficient flowability. In contrast, the failed formulations (PGS, HFEC, and ALM) were increased by 5% to produce the tablets. At this percentage, HFEC provided an acceptable tablet property. However, PGS and ALM formulations still exhibited high tablet friability, owing to the low compressibility profile (value <5). These findings confirmed that the properties of the powder mixture could demonstrate non-linearity. As a result, the prediction of CP value may not be accurate in some formulations [38]. In this case, the failure was found in the formulations in which the excipients exhibited undesirable compressibility (PGS and ALM). Thus, these excipients were not suitable for DC production. However, most formulations showed an acceptable tablet property within 5% variation from the CP value based on the reliability factor of the SeDeM diagram (*f* = 0.952), which was considered as an insignificant difference [39].

Paracetamol was primarily absorbed in the proximal portion of the small intestine, with negligible absorption in the stomach [40]. In this work, the dissolution study was initially conducted in two different mediums, i.e., hydrochloric acid pH 1.2 (gastric fluid) and phosphate buffer pH 5.8 (intestinal fluid). The dissolution profile of successful formulations revealed that CS, PMSS, PGS, and HFEC formulations showed cumulative drug release of greater than 80% after 30 min in both mediums (Figure 3A,B), which met the USP requirement [41]. To characterize the behavior of CS along all the gastrointestinal tract, additional dissolution tests in pH 4.5 and pH 6.8 mediums were carried out. The result showed that the difference in the pH of dissolution mediums did not significantly affect the drug release profile of CS formulation (Figure 3C). For the intestinal fluid condition, the excipients which were co-processed with a disintegrant (i.e., CS and HFEC) yielded faster drug release than others. The CS formulation exhibited the fastest and the greatest drug release (>80% within 5 min). The lubricant co-process of HFEC limited water penetration into HFEC tablets, thus causing slight prolongation of HFEC dissolution. A fast dissolution profile was also observed for the PGS formulation but could be due to the effect of low tensile strength [42]. On the other hand, the dissolution profile of SMCC and ALM formulations failed to meet the requirement. This result suggested that the strong bonding of SMCC tablets limited the tablet dissolution [43], while low disintegration inhibited the dissolution of ALM tablets. The difference in the pH of dissolution mediums did not significantly affect the drug release profile of CS, SMCC, and ALM formulations, while the low pH condition slightly improved drug release from the PMSS and PGS formulations. In contrast, the HFEC formulation showed a slower drug release in acidic medium condition due to the decrease in swellability of MCC in acidic medium [44].

### 2.5. Validation of CS Functionality

To validate the functionality and capability of CS, ibuprofen was selected as an additional model drug. Ibuprofen generally exhibits poor flow and poor compressibility, as a result of its high powder cohesiveness and viscoelasticity [45]. The SeDeM diagram of ibuprofen revealed that the flowability and the compressibility were lower than the acceptable value, suggesting that it was not suitable for DC (Table 6). To compensate for the undesirable properties of ibuprofen, 12% and 40% of CS were required in the mixture to adjust the compressibility and the flowability, respectively. The higher value, 40%, was selected for the formulation. A mixture of ibuprofen (60%) and CS (40%) was prepared according to the calculated CP value. The SeDeM-ODT diagram showed that the mixture could be produced with DC as the IPP and IGC values were higher than 5 (Table 6). Moreover, the IGCB values surpassed the acceptable value, indicating that this formulation did not require an additional disintegrant. However, the below-optimum compressibility parameter of the mixture (2.58) did suggest a potential defect upon tableting. Indeed, the tablets fabricated from the mixture failed the friability test (Table 7), likely caused by poor compressibility, despite having all other tablet properties (weight variation, disintegration, and drug content) in compliance with the specifications [46]. According to the reliability factor of the SeDeM expert system, the concentration of CS was increased by 5% at a time from the calculated CP value until the tablet properties met the requirements. In this case, the CS concentration was increased to 50% (formulation 2), to obtain ibuprofen tablets with acceptable friability, while it maintained the weight variation and drug content within the specifications. This tablet formulation also exhibited good disintegration, resulting in a desirable dissolution profile. More than 80% of the ibuprofen was released within 60 min, which complied with the specification [46].

### 2.6. Stability Study

The successful CS formulation was employed in the stability study under accelerated conditions at 40 ± 2 °C and 75 ± 5% RH for 45 days. The samples were collected at 0, 30, and 45 days to analyze tablet characteristics. The results revealed that no significant change was observed in the sample from each sampling time (Table 8). The weight variation, friability, and drug content were complied with the USP specifications for paracetamol tablets. The dissolution profile showed that the cumulative drug release was higher than 80% for all samples. In addition, the difference factor (f_1_) and similarity factor (f_2_) were within the specifications (0–15 for f_1_ factor and 50–100 for f_2_ factor), which indicated the sameness of the dissolution profile after 30 and 45 days of storage [47].

## 3. Materials and Methods

### 3.1. Materials

Native rice starch (RS) (Lot no. 709161) was purchased from Thai Flour Industry Co., Ltd. (Bangkok, Thailand). Monochloroacetic acid (MCA, CAS No. 79-11-8, Product Code 8004121000) and sodium silicate (CAS No. 1344-09-8, Product Code 1056212500) were purchased from Merck (Hohenbrunn, Germany). Epichlorohydrin (ECH, CAS No. 106-89-8, Product Code E1055) was supplied by Sigma-Aldrich (Steinhiem, Germany). Agglomerated lactose monohydrate (ALM, Tablettose^®^ 80) (Product code: L104314615A552) was the product of Meggle pharma (Wasserburg, Germany). Silicified microcrystalline cellulose (SMCC 90, Prosolv^®^) (Product code: P9D8L19) and high-functionality excipient composite (HFEC, Prosolv EasyTab^®^ SP) (Product code: 6809074049) were from JRS Pharma, Rosenberg, Germany). Paracetamol (Lot no. 01610131) was purchased from Vittayasom Sriracha Co., Ltd. (Chonburi, Thailand). Carboxymethyl rice starch cross-linked with ECH (CCMS) was produced using the method and conditions described in the previous study [11]. The reaction time ratio of etherification and cross-linking was selected based on the disintegration property at 1:0.67.

CS was prepared as reported in the previous study [9]. In brief, CCMS (10 g) was dispersed in distilled water, and stirred continuously until completely swelled. Then, RS (100 g) and sodium silicate solution (11 mL) were added and the mixture was homogenized. The co-processed particle was produced by spray drying of the mixture, using a B-290 mini spray dryer (Buchi, Flawil, Switzerland) with a 2.0 mm nozzle tip. The spray dry conditions were set at 190 °C, 95% of the aspirator, and 18% of the pump. A physical mixture (PMSS) was prepared by blending RS (88.5%), CCMS (8.9%), and spray dried sodium silicate (2.6%) in a plastic bag for 15 min. Pregelatinized rice starch (PGS) was produced by dispersing RS in distilled water (40% *w*/*v*). Starch slurry was heated at 60 °C for 10 min and dried in a hot air oven at 55 °C for 24 h.

### 3.2. Optimization of Excipients and APIs Powder Using SeDeM Expert System

The SeDeM-ODT Expert System evaluated six factors, which were derived from 15 parameters pertinent to the physical characteristics and pharmaceutical functionality of the materials in order to investigate powder characteristics for DC manufacturing (Table 9). These parameters were determined for excipients and API using the following methodologies, which were repeated at least in triplicate.

#### 3.2.1. Dimension

Bulk density (Da)

Bulk density is defined as the powder mass divided by the loosely packed powder volume. It was measured according to European Pharmacopeia (2.9.34). A known mass (g) of powder was poured into a 100 mL graduated cylinder (readable to 1 mL) [48]. The appearance of bulk volume was read, and then bulk density was calculated using Equation (1):Da = M/Va (1)
where Da is the bulk density, M is the powder mass (g), and Va is the bulk volume (mL).

Tapped density (Dc)

Tapped density is the powder mass divided by the powder volume after continuous tapping to a constant value. It was measured according to European Pharmacopeia (2.9.34) by mechanical tapping using a Jolting volumeter (Stav 2003, Erweka, Langen, Germany) of a 100 mL graduated cylinder (readable to 1 mL) containing the sample powder [48]. The sample powder was continuously tapped and the volume was read after 10, 500, and 1250 taps. When the difference between V_500_ and V_1250_ was less than 2 mL, the tapped volume was read as V_1250_. Then, tapped density was calculated using Equation (2):Dc = M/Vc(2)
where Dc is tapped density, M is powder mass (g), and Vc is tapped volume (mL).

#### 3.2.2. Compressibility

Interparticle porosity (Ie)

The interparticle porosity represents the pore space between the particles. It was calculated from the bulk and tapped density values using Equation (3):Ie = (Dc − Da)/(Dc × Da)(3)

Carr index (% IC)

The Carr index is determined by the difference between the bulk and tapped densities. It measures the ability of powder to interact with other particles and resist powder flow. Thus, it indicates powder compressibility. The Carr index was calculated using Equation (4):% IC = (Dc − Da/Dc) × 100(4)

Cohesion index (ICD)

The cohesive index represents the compressibility of materials. The powder (500 mg) was compressed into a tablet with a 11.0-mm flat-face punch using a hydraulic press machine (C, Carver, Wabash, IN, USA) at 1.0 ton of compression force. In the case of uncompressible powder, 3.50% of standard lubricant was added, which included talc 2.36%, colloidal silicon dioxide 0.14%, and magnesium stearate 1.00%, and mixed for 2 min before compression. The tablet crushing strength (N) was determined using a tablet hardness tester (PTB-311, Pharmatest, Hainburg, Germany). This value indicated the maximum compressive stress that can be tolerated in the tablet without fracture.

#### 3.2.3. Flowability

Hausner ratio (IH)

The Hausner ratio indirectly indicates powder flowability. It represents powder resistance against flow due to the particle interaction, which is based on powder density, size and shape, moisture content, and powder cohesiveness. It is calculated using bulk and tapped densities values using Equation (5):IH = Dc/Da(5)

Angle of repose (α)

The angle of repose is related to interparticular friction that limited particle movement. It is determined by a drained angle of repose method according to European Pharmacopeia (2.9.36) [49]. The powder was poured through a funnel that was fixed with a stand and set 10 cm above the table surface [24]. The height and the radius of the base of the powder bulk were measured, and then the angle of repose was calculated using Equation (6):Ө = tan^−1^ (h/r)(6)
where Ө is the angle of repose (°), h is the height of the sample cone, and r is the radius of the base of the sample bulk.

Powder flow (t″)

This factor determines the rate of powder flow through an orifice, which varies depending on particle morphology or process. Powder flow was conducted according to European Pharmacopeia (2.9.16) using a typical apparatus, where a funnel was fixed to a stand and set at 10 cm above the table surface [50]. The powder was poured steadily through a funnel. Powder flow was expressed in second related to 100 g of sample powder.

#### 3.2.4. Stability

Loss on drying (%HR)

This factor determines the percentage of powder which can be volatilized under tested conditions. It was measured using a moisture analyzer (MX-50, A&D, Tokyo, Japan). Approximately one gram of powder was accurately weighed into the sample pan. Then, the sample was heated at 105 °C to constant weight and the percentage of weight loss was measured.

Hygroscopicity (%H)

Hygroscopicity represents the ability of the powder to adsorb or absorb water from the environment. This parameter was determined at 76 ± 2% relative humidity (RH) at room temperature for 24 h. Powder (250 mg) was weighed into a 2.5 cm diameter, pre-weighed cup. The cups were placed in the tight containers which housed saturated salt solutions at the studied RH. After 24 h, the cups were re-weighed and the percentage of the weight gained was calculated.

#### 3.2.5. Dosage

Particle size (%Pf)

This parameter represents the percentage of particles that passed through a 50 um sieve. The test was conducted using a sieve vibrator (AS 200 control, Retsch, Haan, Germany) at 1.0 g for 10 min.

Homogeneity Index (Iθ)

The Iθ value indicates the homogeneity of the particle size distribution of powder. This parameter was measured using a sieve vibrator with four different sieve sizes (45, 106, 212, and 355 µm). The powder sample (20 g) was determined with agitation at 1.0 g for 10 min. The Iθ value was calculated using Equation (7):Iθ = Fm/[100 + (dm − dm − 1)Fm − 1 + (dm + 1 − dm)Fm + 1 + (dm − dm − 2)Fm − 2 + (dm + 2-dm) Fm + 2 … + (dm − dm − n)Fm − n+ (dm + n − dm)Fm + n](7)
where Iθ is the relative homogeneity index, Fm, Fm + 1, and Fm − 1 are particle percentages remaining in the majority range, above the majority range, and below the majority range, respectively, dm, dm + 1, and dm − 1 are the mean diameter of particle in the majority range, above the majority range, and below the majority range, respectively, and n is the order number of the fraction studied under a series, relating to the major fraction.

#### 3.2.6. Disgregability

The tablets were prepared by compression of the excipient powder (500 mg) with a 11.0-mm flat-face punch using a hydraulic press machine (C, Carver, Wabash, IN, USA) at 1.0 ton of compression force. The surface of punch and die were pre-lubricated with magnesium stearate solution (5% *w*/*v*) before compression.

Effervescence test (DE)

This test determines the possibility of a tablet to be bucodispersible. The tablet (500 mg) was placed in a beaker containing an excess amount of distilled water at room temperature. At the point which the tablet completely disaggregated, the time was recorded (min).

Disintegration time with disk (DCD)

This test imitates the mechanical movement inside the mount during the taking of a tablet, which promoted tablet disintegration. The test was conducted according to the standard USP method [51]. The medium temperature was controlled at 37 ± 0.5 °C. The tablets (500 mg) were placed in a basket and covered with a disk, then tablet disintegration time was observed and recorded after the tablet completely disintegrated. Six tablets were determined simultaneously.

Disintegration time without disk (DSD)

This test is specific for investigation of bucodispersible tablets. The test was conducted according to the standard USP method described for measuring DCD, except the disk was removed from the test. The tablet disintegration time was observed and recorded after the tablet disintegrated completely. Six tablets were determined simultaneously.

### 3.3. Conversion of Parameter Values to Radius Values of the SeDeM-ODT Diagram

The values of the parameters (v) obtained from topic 2.2 were calculated to convert into the radius values (r) of the SeDeM-ODT diagram (Table 9). The SeDeM-ODT diagram, which consisted of 15 polygons of each parameter, was then plotted. The radius scale ranged from 0–10; a value of 5 was considered the minimum acceptable value of each parameter. The parameters were integrated into six incidence factors according to Table 1. The incidence factor values were calculated as a mean of their respective integrated parameters.

### 3.4. The Determination of the Index Using SeDeM-ODT Diagram

The index parameter (IP), index profile parameter (IPP), index of good compressibility (IGC), and index of good compressibility and bucodispersibility (IGCB) were calculated to evaluate the suitability of powder for applying in DC manufacturing using the following equations:IP = n° P > 5/n° Pt(8)
where n° P > 5 is the number of parameters which radius values > 5, while n° Pt is the number of total studied parameters. The acceptability limit of IP index should correspond to >0.5.
IPP = Average radius values of all parameters(9)
IGC or IGCB = IPP × f(10)
*f**=* Polygonal area/circle area(11)

The IGC was calculated based on 12 parameters, excluding the disgregability factors. The IGCB was obtained based on 15 parameters, including the three disgregability parameters. The acceptability limit of IPP, IGC, and IGCB should correspond to 5 or higher.

### 3.5. Calculation of the Number of Excipients Required to Adjust API Characteristics for DC Formulation

The flowability and compressibility of the API were corrected by combining it with a suitable excipient. The number of excipients required to adjust API characteristics was calculated using Equation (12), based on the SeDeM-ODT diagram. All factors can be adjusted using Equation (12). However, compressibility and flowability were given priority, as they represented the major important factors for DC production:CP = 100 − [((RE − R)/(RE − RP)) × 100](12)
where CP is the percentage of a corrective excipient, RE is the average radius value of the corrective excipient, RP is the average radius value of the API to be corrected, and R represents the required average radius value of the mixture of excipient and API; thus, 5 is the minimum value allowed to correct API characteristics.

### 3.6. Formulation Study

#### 3.6.1. Tablet Preparation

Tablets were prepared using a direct compression method. Paracetamol powder was homogenously blended with excipients using a geometric dilution technique for 15 min. After that, magnesium stearate (1%) was added and mixed for 2 min. The powder mixture (100 g) was compressed into tablets using a single punch tableting machine (CMT 12, Charatchai, Bangkok, Thailand), housed with a 10.3 mm round, flat-face punch, at a tableting speed of 2640 tab/h. The target weight of a tablet was set at 300 mg.

#### 3.6.2. Evaluation of Tablet Properties

Weight variation was determined, according to the USP method [52], by weighing and recording the weight of 30 individual tablets. Tablet hardness was measured using a tablet hardness tester and conducted on 10 tablets. The friability test was conducted according to the USP method. Approximately 6.5 g of the tablets was sampled and dedusted before the test [53]. The tablets were accurately weighed and placed in a friability tester. The test was conducted on the rotation at 25 rpm for 4 min. The tested tablets were dedusted and re-weighed. Tablet friability was calculated from the weight difference. The acceptable friability of tablets should be less than 1.0%. The disintegration test was determined in at least six tablets using a basket apparatus according to the USP method [51]. Each tablet was placed in a basket tube and covered with a disc. The test was carried out in a distilled water medium, maintained at a temperature of 37 ± 0.5 °C throughout the test. The disintegration time was recorded after the entire tablet completely disintegrated. The drug content percentage (DCP) of paracetamol tablet was determined according to the BP method, which should be in the range of 95.0–105.0% of the stated amount [54].

#### 3.6.3. Tablet Criteria

Tablet formulation was determined as a successful formulation based on three parameters, including weight variation, friability (%), and drug content (%). These parameters reflected the ability of the powder blend to fill the dies during tablet compression and provided a suitable tablet strength during the production and throughout its shelf life. DCP represented the uniformity of the powder blend. To be classified as a successful formulation, a tablet formulation must meet all the requirements. After that, the excipient percentage in the formulation was increased by 5% for the failed formulations or decreased by 5% for the successful formulations based on the reliability factor of SeDeM diagram (f = 0.952). The paracetamol tablets were formulated, and the tablet properties were evaluated using the procedure described above.

#### 3.6.4. Dissolution Study

The successful formulations were selected and subjected to dissolution tests using a paddle apparatus according to the standard USP method [55]. For PGS and ALM formulations, the one that contained a higher number of excipients was selected to enter the test. The test was carried out for 30 min, using different dissolution mediums (hydrochloric acid pH 1.2 and phosphate buffer pH 5.8), at a controlled temperature of 37 ± 0.5 °C throughout the test, where the paddle speed was set at 50 rpm. CS formulation was also conducted with additional dissolution mediums (acetate buffer pH 4.5 and phosphate buffer pH 6.8) to evaluate CS dissolution behavior along the gastrointestinal tract. Five ml of sample was taken from the vessel at 2, 5, 10, 15, 20, and 30 min, with equal volume replacement of fresh medium for each sampling. The samples were analyzed using a UV-Vis 2450 spectrophotometer (Shimadzu, Japan) at 243 nm. The cumulative drug release percentage was then calculated. The test was repeated at least in triplicate.

### 3.7. Validation of CS Functionality

The characteristics of ibuprofen powder (model drug) were investigated using the SeDeM expert system, and the poor characteristics (flowability and/or compressibility) were corrected with CS. The number of excipients required to compensate for ibuprofen characteristics (CP, %) was calculated. The ibuprofen formulation was prepared according to the CP value by DC technique; then, the SeDeM-ODT expert system was applied to determine formulation characteristics. Ibuprofen tablets (300 mg/tablet) were produced using a single punch tableting machine with a 10.3 mm round, flat-face punch. The tablet properties were evaluated as described earlier. After that, CS concentration was varied by a 5% increase from the calculated CP value until the tablet properties met the requirements. The dissolution property of the successful formulation was determined according to the standard USP method.

### 3.8. Stability Study

The successful CS formulation was subjected to stability study under accelerated condition(s) according to the International Conference on Harmonization (ICH) guidelines. The tablets were stored at 40 ± 2 °C and 75 ± 5% RH for 45 days. Stability samples were taken at 0, 30, and 45 days to determine tablet characteristics, which included weight variation, breaking force, friability, disintegration time, drug content, and dissolution study. A difference factor (f_1_) and similarity factor (f_2_) were applied to determine the sameness of the dissolution profile [47]:(13)$$f_{1}=(\sum_{t=1}^{n}|R_{t}-T_{t}|/\sum_{t=1}^{n}R_{t})\times 100$$
(14)$$f_{2}=50\times {\text{log}}_{10}\left[100/\right.\;\sqrt{1+{\left(\sum_{t=1}^{n}(R_{t}-T_{t}\right)}^{2}/n}]$$
where n is the number of sampling time point, Rt is the average of cumulative drug release of day 0 tablets at time t, and Tt represents the average of cumulative drug release of day 30 or day 45 tablets at that similar time point.

### 3.9. Statistics

All tests were conducted at least in triplicate and the data are presented as average values. Statistical analysis was performed using a one-way analysis of variance (ANOVA) in SPSS (version 19.0). Significance tests were analyzed using Tukey’s honestly significant difference (HSD) multiple range test at a 95% confidence level (*p* < 0.05).

## 4. Conclusions

The formulation study of a co-processed, rice starch-based excipient (CS) was evaluated using the SeDeM expert system for application in direct compression (DC) formulation. CS exhibited acceptable powder characteristics based on the values of six incidence factors. The comprehensive indices (IGC and IGCB) of CS surpassed the acceptable values, suggesting that CS was suitable for DC manufacturing without the addition of a disintegrant to the tablet formulation. The CP value of the excipient required to compensate for the property of the model drug paracetamol was selected based on flowability. The formulation study revealed that CS can be diluted up to 60% with an API of poor flow and compressibility in the paracetamol formulation. At this percentage, CS tablets showed narrow tablet weight variation, low friability, suitable drug content, and fast disintegration time. The dissolution study showed that CS can release more than 80% of the drug content in less than 5 min at all tested pH conditions, which was superior to other excipients. CS was comparable to HFEC, and can be used to compensate the inadequate flow and compressibility of APIs. Other excipients, such as SMCC, required the addition of a disintegrant, while PGS and ALM may not be suitable for poorly compressible APIs. PMSS, a simple, physical mixture of various excipients, could not significantly improve the API properties and, thus, it was not considered as a multifunctional excipient. The CS formulated with 50% of ibuprofen showed satisfactory tablet properties, which implied the wide range of CS functionality to formulate with various APIs. The stability study of the CS tablets showed no significant change after 45 days storage. Overall, CS exhibited properties and characteristics as a multifunctional excipient for direct compression of pharmaceutical tablets.

## Figures and Tables

**Figure 1 pharmaceuticals-14-01047-f001:**
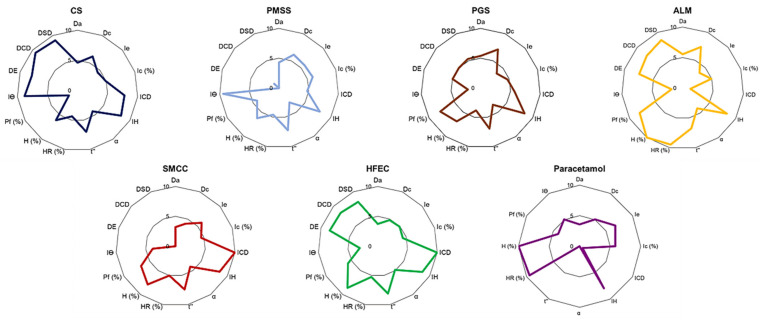
SeDeM-ODT diagrams of CS, PMSS, PGS, and commercial DC excipients, and a SeDeM diagram of paracetamol.

**Figure 2 pharmaceuticals-14-01047-f002:**
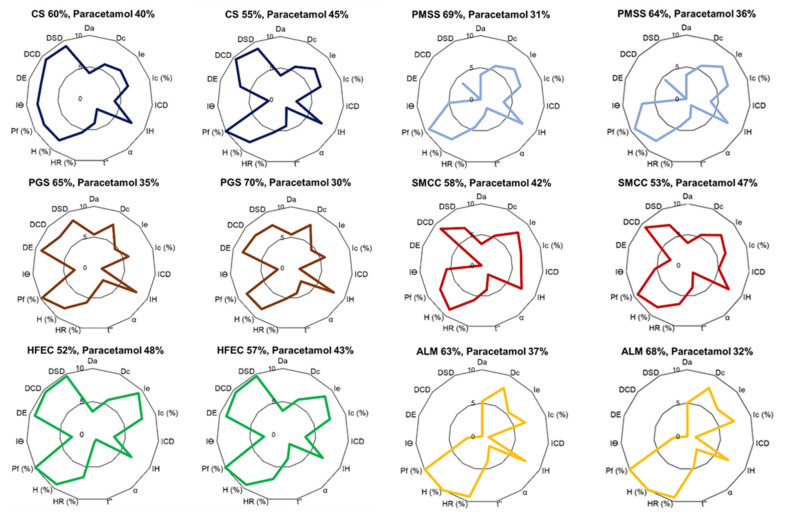
SeDeM-ODT diagrams of paracetamol formulations containing different types and ratios of directly compressed excipients.

**Figure 3 pharmaceuticals-14-01047-f003:**
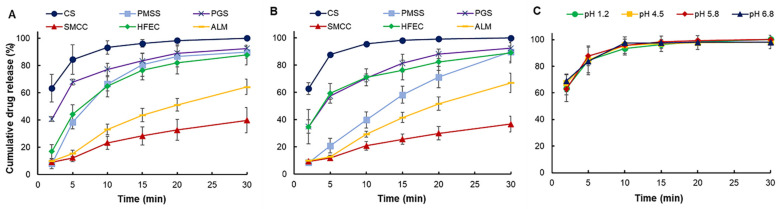
Drug release profiles of paracetamol tablets formulated using various excipients at pH 1.2 (**A**) and 5.8 (**B**), and compiled release profiles of paracetamol-CS formulation at four different pH values along the GI tract (**C**).

**Table 1 pharmaceuticals-14-01047-t001:** Parameter radius values of paracetamol and excipients.

Samples	Da	Dc	Ie	Ic (%)	ICD	IH	α	t″	HR (%)	H (%)	Pf (%)	Iϴ	DE	DCD	DSD
CS	4.62 ^e^	6.14 ^d^	4.48 ^b^	4.96 ^c^	7.83 ^e^	8.35 ^b^	4.27 ^d^	7.02 ^b^	4.37 ^b^	6.09 ^a^	1.65 ^a^	8.82 ^d^	7.97 ^d^	9.34 ^f^	9.07 ^e^
PMSS	4.31 ^d^	6.10 ^d^	5.75 ^d^	5.95 ^d^	4.78 ^c^	7.88 ^a^	3.00 ^b^	7.23 ^b^	4.58 ^bc^	6.25 ^a^	4.21 ^b^	9.40 ^d^	0.00 ^a^	1.08 ^b^	0.00 ^a^
PGS	5.42 ^f^	7.09 ^e^	3.62 ^a^	4.71 ^bc^	5.67 ^d^	8.46 ^bc^	3.45 ^c^	6.96 ^b^	3.83 ^a^	6.88 ^b^	7.83 ^d^	2.18 ^ab^	4.95 ^b^	4.98 ^c^	5.33 ^b^
SMCC	3.20 ^a^	4.14 ^a^	5.90 ^d^	4.53 ^ab^	10.00 ^f^	8.53 ^c^	4.22 ^d^	7.39 ^bc^	5.03 ^cd^	7.92 ^c^	6.66 ^cd^	3.86 ^bc^	0.00 ^a^	0.00 ^a^	0.00 ^a^
HFEC	3.75 ^c^	4.80 ^b^	4.90 ^c^	4.40 ^a^	10.00 ^f^	8.59 ^c^	4.88 ^e^	8.12 ^c^	5.26 ^d^	8.63 ^d^	5.77 ^bc^	3.05 ^ab^	8.46 ^e^	8.38 ^e^	8.12 ^c^
ALM	5.55 ^f^	7.37 ^f^	3.72 ^a^	4.95 ^c^	1.26 ^b^	8.35 ^b^	4.08 ^d^	6.76 ^b^	9.47 ^e^	9.99 ^e^	8.13 ^d^	1.84 ^a^	7.47 ^c^	7.92 ^d^	8.58 ^d^
Paracetamol	3.59 ^b^	5.07 ^c^	6.76 ^e^	5.83 ^d^	0.57 ^a^	7.94 ^a^	0.00 ^a^	0.00 ^a^	9.50 ^e^	9.99 ^e^	4.04 ^b^	5.07 ^c^	N/A	N/A	N/A

A common letter (a–f) shows that the value is not significantly different within group by Tukey HSD test at a 5% level of significance (*p* < 0.05).

**Table 2 pharmaceuticals-14-01047-t002:** Incidence factor and parametric index values of paracetamol and excipients.

Samples	Incidence Factors						SeDeM Diagram			SeDeM-ODT Diagram		
	Dimension	Compressibility	Flowability	Stability	Dosage	Disgregability	IP	IPP	IGC	IP	IPP	IGCB
CS	5.38 ^e^	5.75 ^c^	6.54 ^cd^	5.23 ^a^	5.23 ^ab^	8.79 ^f^	0.58	5.72	5.44	0.67	6.33	6.15
PMSS	5.22 ^d^	5.49 ^c^	6.04 ^b^	5.42 ^a^	6.80 ^b^	0.36 ^b^	0.58	5.79	5.51	0.47	4.70	4.57
PGS	6.25 ^f^	4.67 ^b^	6.29 ^bc^	5.35 ^a^	5.00 ^ab^	5.08 ^c^	0.58	5.51	5.24	0.53	5.42	5.27
SMCC	3.67 ^a^	6.81 ^e^	6.71 ^d^	6.48 ^b^	5.26 ^ab^	0.00 ^a^	0.58	5.95	5.66	0.47	4.76	4.62
HFEC	4.27 ^b^	6.43 ^d^	7.20 ^e^	6.95 ^c^	4.41 ^a^	8.32 ^e^	0.50	6.01	5.72	0.60	6.47	6.29
ALM	6.46 ^g^	3.31 ^a^	6.40 ^c^	9.73 ^d^	4.99 ^ab^	7.99 ^d^	0.67	5.96	5.67	0.73	6.36	6.18
Paracetamol	4.33 ^c^	4.39 ^b^	2.65 ^a^	9.74 ^d^	4.56 ^a^	N/A	0.50	4.86	4.63	N/A	N/A	N/A

A common letter (a–g) shows that the value is not significantly different within group by Tukey HSD test at the 5% level of significance (*p* < 0.05).

**Table 3 pharmaceuticals-14-01047-t003:** Percentage of excipients required to correct paracetamol characteristics in tablet formulations.

Excipients	Compressibility				Flowability					
	CS	PMSS	SMCC	HFEC	CS	PMSS	SMCC	HFEC	ALM	PGS
RE	5.75	5.49	6.81	6.43	6.54	6.04	6.71	7.20	6.40	6.29
RP	4.39	4.39	4.39	4.39	2.65	2.65	2.65	2.65	2.65	2.65
R	5.00	5.00	5.00	5.00	5.00	5.00	5.00	5.00	5.00	5.00
% excipient (CP)	44.86	55.45	25.31	29.98	60.38	69.43	57.88	51.72	62.72	64.59

RE = the average radius value of the corrective excipient; RP = the average radius value of the API to be corrected; CP (%) = the percentage of a corrective excipient; R = the required average radius value.

**Table 4 pharmaceuticals-14-01047-t004:** Incidence factor and parametric index values of paracetamol tablets formulated with various types and ratios of excipients.

Formulations	Excipient (%)	Paracetamol (%)	Incidence Factor						SeDeM			SeDeM-ODT		
			Dimension	Compressibility	Flowability	Stability	Dosage	Disgregability	IP	IPP	IGC	IP	IPP	IGCB
CS	60	40	4.97 ^f^	5.68 ^d^	4.55 ^ef^	6.81 ^b^	8.26 ^e^	8.78 ^h^	0.67	5.90	5.62	0.67	6.48	6.29
CS	55	45	4.75 ^de^	5.48 ^d^	4.45 ^def^	6.78 ^b^	5.96 ^ab^	8.33 ^g^	0.58	5.40	5.14	0.67	5.99	5.81
PMSS	69	31	4.82 ^e^	5.78 ^de^	3.94 ^bc^	6.07 ^a^	8.17 ^de^	1.25 ^b^	0.58	5.60	5.33	0.47	4.73	4.60
PMSS	64	36	4.73 ^cde^	6.04 ^ef^	3.88 ^b^	6.18 ^a^	8.10 ^cde^	1.46 ^c^	0.58	5.65	5.38	0.47	4.81	4.67
PGS	65	35	6.52 ^gh^	3.90 ^a^	4.55 ^ef^	6.95 ^bc^	6.45 ^abcd^	8.28 ^g^	0.58	5.43	5.17	0.67	6.00	5.13
PGS	70	30	6.60 ^h^	3.82 ^a^	4.96 ^g^	6.75 ^b^	5.33 ^a^	7.97 ^f^	0.58	5.31	5.05	0.67	5.84	5.25
SMCC	58	42	4.38 ^a^	7.05 ^i^	4.41 ^de^	7.25 ^cd^	6.71 ^abcde^	5.17 ^d^	0.75	5.92	5.64	0.67	5.77	5.60
SMCC	53	47	4.53 ^b^	6.26 ^fg^	4.25 ^cde^	7.35 ^d^	7.33 ^bcde^	6.43 ^e^	0.75	5.83	5.55	0.73	5.95	5.78
HFEC	52	48	4.63 ^bc^	6.84 ^hi^	2.97 ^a^	7.94 ^e^	6.57 ^abcde^	9.49 ^i^	0.58	5.64	5.37	0.67	6.41	6.23
HFEC	57	43	4.71 ^cd^	6.54 ^gh^	4.18 ^bcd^	8.11 ^e^	6.41 ^abc^	9.38 ^i^	0.58	5.89	5.60	0.67	6.58	6.39
ALM	63	37	6.61 ^h^	4.36 ^b^	4.38 ^de^	9.63 ^f^	6.08 ^ab^	0.00 ^a^	0.67	5.91	5.62	0.53	4.72	4.59
ALM	68	32	6.50 ^g^	4.91 ^c^	4.73 ^fg^	9.68 ^f^	6.07 ^ab^	0.00 ^a^	0.75	6.12	5.83	0.60	4.90	4.75

A common letter (a–i) is not significantly different within group by Tukey HSD test at the 5% level of significance (*p* < 0.05). IP = index parameter; IPP = index profile parameter; IGC = index of good compressibility; IGCB = index of good compressibility and bucodispersibility.

**Table 5 pharmaceuticals-14-01047-t005:** Properties of paracetamol tablets formulated with various types and ratios of excipients.

Formulations	Excipient (%)	Paracetamol (%)	Tablet Weight (mg)	Breaking Force (Kp)	Friability (%)	Disintegration Time (s)	Drug Content (%)	Verdict	Reason for Failure
CS	60	40	299.57 ± 6.02	8.4 ± 1.7	0.69 ± 0.13	11.00 ± 1.10	102.02 ± 1.48	Success	
CS	55	45	295.91 ± 7.00	5.0 ± 1.0	1.44 ± 0.51	11.83 ± 1.60	100.94 ± 2.61	Failure	Friability
PMSS	69	31	294.41 ± 8.08	6.9 ± 1.9	0.51 ± 0.11	74.33 ± 3.50	98.45 ± 0.57	Success	
PMSS	64	36	300.09 ± 22.24	6.1 ± 0.7	0.38 ± 0.11	74.17 ± 3.31	104.87 ± 1.93	Failure	High weigh variation
PGS	65	35	271.49 ± 17.51	2.4 ± 0.5	34.37 ± 4.28	44.00 ± 6.48	93.40 ± 3.38	Failure	Friability, high weight variation, and low drug content
PGS	70	30	290.56 ± 7.59	3.4 ± 0.9	18.95 ± 3.00	33.83 ± 6.11	103.60 ± 5.61	Failure	Friability
SMCC	58	42	295.22 ± 2.48	11.5 ± 0.8	0.38 ± 0.03	86.17 ± 9.17	103.63 ± 0.76	Success	
SMCC	53	47	278.10 ± 16.33	7.0 ± 1.1	0.28 ± 0.03	42.67 ± 9.00	99.31 ± 4.22	Failure	High weigh variation
HFEC	52	48	289.36 ± 11.93	4.3 ± 0.5	1.86 ± 0.77	5.67 ± 1.03	96.60 ± 0.47	Failure	Friability
HFEC	57	43	297.30 ± 5.64	6.9 ± 1.4	0.48 ± 0.10	8.67 ± 1.51	101.34 ± 1.31	Success	
ALM	63	37	294.82 ± 33.18	1.8 ± 0.2	47.95 ± 1.96	899.83 ± 55	91.46 ± 2.02	Failure	Friability, high weight variation, low drug content
ALM	68	32	289.30 ± 7.64	1.9 ± 0.3	33.63 ± 3.00	618.67 ± 14.85	98.77 ± 1.70	Failure	Friability

**Table 6 pharmaceuticals-14-01047-t006:** Incidence factor and parametric index values of the ibuprofen powder and mixture.

Samples	Incidence Factor						SeDeM			SeDeM-ODT		
	Dimension	Compressibility	Flowability	Stability	Dosage	Disgregability	IP	IPP	IGC	IP	IPP	IGCB
Powder	4.90	4.89	3.95	9.68	9.98	N/A	0.67	6.30	6.00	N/A	N/A	N/A
Mixture	5.52	2.58	6.93	7.76	7.95	7.16	0.67	5.92	5.63	0.67	6.16	5.99

IP = index parameter; IPP = index profile parameter; IGC = index of good compressibility; IGCB = index of good compressibility and bucodispersibility.

**Table 7 pharmaceuticals-14-01047-t007:** Properties of ibuprofen tablets formulated with different percentages of CS.

Formulations	CS (%)	Ibuprofen (%)	Tablet Weight (mg)	Breaking Force (Kp)	Friability (%)	Disintegration Time (s)	Drug Content (%)	Verdict	Reason for Failure
1	40	60	299.42 ± 5.27	3.2 ± 0.5	2.27 ± 0.32	21.33 ± 2.34	97.75 ± 2.57	Failure	High friability
2	50	50	301.96 ± 3.13	4.5 ± 0.3	0.90 ± 0.09	34.17 ± 2.32	102.45 ± 1.92	Success	

**Table 8 pharmaceuticals-14-01047-t008:** Properties of paracetamol tablets formulated with CS at different days.

Tests	CS Formulation		
	Day 0	Day 30	Day 45
Tablet weight (mg)	299.57 ± 6.02	295.58 ± 2.80	301.81 ± 5.26
Breaking force (Kp)	8.4 ± 1.7	8.2 ± 2.2	8.2 ± 0.9
Friability (%)	0.69 ± 0.13	0.64 ± 0.01	0.60 ± 0.12
Disintegration time (s)	11.00 ± 1.10	16.33 ± 2.94	17.00 ± 3.35
Drug content (%)	102.02 ± 1.48	101.27 ± 1.10	101.94 ± 2.00
Cumulative drug release (%)	>80%	>80%	>80%
Difference factor (f_1_)	N/A	3.24	1.85
Similarity factor (f_2_)	N/A	71.02	72.54
Verdict	Success	Success	Success

**Table 9 pharmaceuticals-14-01047-t009:** Evaluated parameters and conversion of parameter values to radius values.

Incidence factor	Parameter	Unit	Limit Value (v)	Radius (r)
Dimension	Bulk density (Da)	g/mL	0–1	10v
	Tapped density (Dc)	g/mL	0–1	10v
Compressibility	Inter-particle porosity (Ie)	-	0–1.2	10v/1.2
	Carr Index (IC)	%	0–50	v/5
	Cohesive Index (Icd)	N	0–200	v/20
Flowability	Hausner ratio (IH)	-	1–3	(30–10v)/2
	Angle of repose (α)	°	0–50	10–(v/5)
	Powder flow (*t*’’)	S	0–20	10–(v/2)
Stability	Loss on drying (%HR)	%	0–10	10–v
	Hygroscopicity (%H)	%	0–20	10–(v/2)
Dosage	Particle < 50 (%Pf)	%	0–50	10–(v/5)
	Homogeneity index (Iϴ)	-	0.2 × 10^−^^2^	500v
Disgregability	Effervescence (DE)	min	0–5	(5-v)2
	Disintegration time with disk (DCD)	min	0–3	(3-v)3.33
	Disintegration time without disk (DSD)	min	0–3	(3-v)3.33

## Data Availability

Data is contained within the article.

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
