# Peer review of "Formulation Study of a Co-Processed, Rice Starch-Based, All-in-One Excipient for Direct Compression Using the SeDeM-ODT Expert System"

_pharmaceuticals, 2021, doi:10.3390/ph14101047_

Round 1

Reviewer 1 Report

In my opinion more dissolution test at different pHs must be added even althouht paracetamol is recommended to be taken after meals, and it is primarily absorbed along the small intestine. In order to characterize the new excipient behavior more conditions must be assayed in order to know the behaviour along all the digestive tract

Author Response

Reviewer 1: In my opinion more dissolution test at different pHs must be added even althouht paracetamol is recommended to be taken after meals, and it is primarily absorbed along the small intestine. In order to characterize the new excipient behavior more conditions must be assayed in order to know the behaviour along all the digestive tract.

Response: Additional dissolution tests were carried out at pH 4.5 and 6.8 for paracetamol-CS tablet formulation. The results are presented in Figure 3C and are also reported/discussed in the text (highlighted RED) of the revised manuscript.

Reviewer 2 Report

The authors have addressed my concerns.

Author Response

Thank you very much.

This manuscript is a resubmission of an earlier submission. The following is a list of the peer review reports and author responses from that submission.

Round 1

Reviewer 1 Report

Dear Authors,

I found your work very detailed and informative. Please discuss whether formulation prepared using CS will remain stable (Physically and chemically) over shelf-life?  

Reviewer 2 Report

Paper is interesting. It is well-designed and well-performed

I like the paper because of the applicability in pharmaceutic industry

Major points

The dissolution study was carried out for 30 min, using phosphate buffer solution (pH 5.8) as a 535medium, at a controlled temperature of 37±0.5 °C throughout the test.  I think it is necessary include dissolution profiles at pH 1,2, 4,5 and 6,8 that are the standard dissolution conditions. 

I think it is necessary to include more molecules than only paracetamol in order to validate the excipient

Minor comments: include the complete name of the abbreviations in figure legends. It is hard to follow the large quantity of abbreviations

Reviewer 3 Report

The authors produced a co-processed excipient containing a new rice starch as a continuation of their previous work (Ref 9.), and in this manuscript, the applicability of this new excipient for the ODT formula to the SeDeM-ODT expert system was investigated. Paracetamol was used as a model drug. The figures and tables are informative.

Some corrections are necessary:. 

-Review the spelling of the manuscript again because there are many spelling errors in the text (eg line 21: repid instead of rapid; line 508 contains friabilitor drum instead of friability drum.) 

-The authors wrote the materials and methods section as Chapter 3, and Chapter 2 presents the results and discussion. The second chapter contains several abbreviations, which are solved only in Chapter 3 in many cases (eg ALM, PGS, etc.). It would be good to have a list of abbreviations at the beginning of the manuscript.

-The used materials do not include CCMS and sodium silicate. For tableting excipients, the brand name and product code are given, but it is not possible to identify the excipient used. Meggle Company markets 3 types of agglomerated lactose monohydrate (Tablettose 70; 80; and 100). This is also the case for Prosolv. 

-3.2.11. the amplitude is not indicated in the section. 

- It would be providential if Part 3.2 were structured according to Table 6 and this would appear in the description. Readers may be confused by the fact that several disintegration time studies are included in the methods section (3.2.14; 3.2.15 and 3.6.1). It would be beneficial if the authors also indicated in the titles of the subsections that they are part of the Disgregability study. In these cases, i.e. 3.2.14. and 3.2.15, how were the tablets prepared under which operating parameters? Did it contain only the excipient? What is the hardness of these tablets, as this is also decisive in the disintegration test? 

- Batch size is missing for 3.6.1. Compression speed (tablets/hour) is not indicated. What compression force was used for the different batches? 

- The number of parallel measurements during the dissolution test is missing.